
# Superintegrable cellular automata and dual unitary gates from Yang-Baxter maps

Tamás Gombor and Balázs Pozsgay

MTA-ELTE "Momentum" Integrable Quantum Dynamics Research Group,
Department of Theoretical Physics, Eötvös Loránd University

## Abstract

We consider one dimensional block cellular automata, where the local update rules are given by Yang-Baxter maps, which are set theoretical solutions of the Yang-Baxter equations. We show that such systems are superintegrable: they possess an exponentially large set of conserved local charges, such that the charge densities propagate ballistically on the chain. For these quantities we observe a complete absence of "operator spreading". In addition, the models can also have other local charges which are conserved only additively. We discuss concrete models up to local dimensions $N \leq 4$, and show that they give rise to rich physical behaviour, including non-trivial scattering of particles and the coexistence of ballistic and diffusive transport. We find that the local update rules are classical versions of the "dual unitary gates" if the Yang-Baxter maps are non-degenerate. We discuss consequences of dual unitarity, and we also discuss a family of dual unitary gates obtained by a non-integrable quantum mechanical deformation of the Yang-Baxter maps.



# 1   Introduction

The dynamics of many body interacting models is generally intractable with analytic methods, and this holds true for both classical and quantum models. It is only in specific circumstances that exact solutions can be found, which makes such systems valuable from a theoretical point of view. One dimensional integrable models form a well studied class of exactly solvable models, having distinctive features such as the existence of a large set of conserved charges and completely elastic and factorized scattering of quasi-particles [1–3]. However, exact solutions can also be found in other circumstances, for example in quantum circuits with dual unitary gates, as we discuss below.

In this paper we consider two special types of dynamical systems: block cellular automata (BCA) and closely related brickwork quantum circuits (also called quantum cellular automata, QCA) in one space dimension. Even though these are relatively simple systems, it is believed that they display many of the important physical features of more generic systems. For recent reviews of QCA see [4, 5].

We focus on integrable cases in both the classical and quantum setting, and consider models with a specific brickwork structure for the local update rules. Systems of this type have been studied recently as simple models for non-equilibrium dynamics, both in the classical and quantum settings (see for example [6, 7]; other relevant references will be given later in the text). However, the understanding of integrable cellular automata with finite configuration spaces is not yet satisfying.

Closely related systems, such as classical integrable equations on discrete space-time lattices are well studied [8–11], and most of the research deals with models with continuous local configuration spaces, such as the complex numbers or some group manifolds. These systems display the hallmarks of integrability, such as the existence of a set of higher charges and commuting flows (the latter is also known as multi-dimensional consistency or the cube condition), the appearance of Lax matrices and the Yang-Baxter equation in various forms [12–14], and also a constrained algebraic growth of the iterated functions [15]. A new algebraic approach to discrete time integrable models was formulated recently in [16].

In contrast, less is known in those cases when the configuration space is also finite; these models are sometimes called "ultra-discrete" [17]. In specific cases they were well studied, see for example the so-called box-ball systems [17], which can be understood as the ultra-

discretization of integrable field equations. Cellular automata can be viewed as a classical dynamical systems over finite fields, and algebraic and algebro-geometric aspects were studied in [18–22].

BCA with Floquet type update rules attracted considerable attention in the last couple of years, both with finite and continuous configuration spaces [23, 24]. In the finite case the most studied system has been the so-called Rule54 model [7, 8, 25], where exact solutions were found for the dynamics despite the model being interacting [26–29]. Quite surprisingly, despite having perhaps the simplest dynamics among interacting models, the algebraic integrability of the Rule54 model is still not clarified, see [30, 31]. A new algebraic framework for spin chains and QCA with "medium range interaction" was developed in [31], where it was shown that the closely related Rule150 model is Yang-Baxter integrable with three-site interactions. This new approach led to other integrable block cellular automata, for example the model of [32] which is the classical version of the folded XXZ model [33, 34].

These recent developments motivate further research on BCA. There are two main questions. First of all, how can we find and classify all integrable models of this sort, and second, which one of these models is interesting and useful from a physical point of view. In this second respect we mention the paper [6] and subsequent works [35, 36] which showed that even a relatively simple solvable system can serve as a useful toy model for generic physical behaviour, such as diffusive transport.

In this paper we consider classical BCA where the update rule is given by a so-called Yang-Baxter map: a set theoretical solution of the Yang-Baxter equation, without any spectral parameters. These maps were introduced by Drinfeld in [37] and studied afterwards in detail in the seminal paper [38]. By now the study of Yang-Baxter maps on finite sets grew into a separate topic in mathematics and mathematical physics. We do not attempt to give a review of this field, instead we refer the reader to the recent work [39] which deals with the enumeration of Yang-Baxter maps up to size $N = 10$.

The novelty of our work is that we build BCA from Yang-Baxter maps; apparently this simple construction was not yet explored in the literature. Closely related constructions were studied in [40, 41], but these works treated translationally invariant quantum spin chains and not the BCA. The model of [6] belongs to the class we are considering, and it is in fact the simplest non-trivial model in this class, as we show in the main text.

Our aim is to explore the consequences of the Yang-Baxter equation on the dynamics of these models, focusing first on the classical examples. These are treated in Sections 3-6 after the general introduction of the setup in Section 2. We find that the models are super-integrable: they possess an exponentially large set of local conservation laws, and this sets them apart from a generic integrable model. Afterwards we also consider quantum mechanical deformations (Section 7), and explain that the models lose their super-integrability, but remain integrable. We should note that conservation laws in cellular automata were already studied some time ago [42, 43] and also more recently [44], but these works treat the usual CA, and not the integrable BCA. Conserved operators in Clifford quantum CA were investigated in [45].

Studying the classification of the Yang-Baxter maps we immediately encounter the property of non-degeneracy [38]. This can be seen as the classical analog of the "dual unitary" property of two-site quantum gates. Therefore we also explore the overlap between the dual unitary and integrable quantum circuit models. The dual unitary circuits are quantum BCA, where the two-site gate is a unitary operator when viewed as a generator of translations in both the time and space directions [46–48]. They are solvable models of quantum computation, which can display integrable and chaotic behaviour as well. A complete parameterization for dual unitary gates is not known beyond local dimension $N = 2$, but quite general constructions were published, see for example [49] or formula (51) of [50] (which was suggested by one of the present authors). In this work we also contribute to the theory of dual unitary models, by

establishing a connection with the Yang-Baxter maps in the integrable cases, and by presenting a non-integrable deformation of such gates, thus obtaining a more general family of dual unitary gates (see Section 8).

In Section 9 we present our conclusions and some open problems.

## 2 Models

In this work we consider classical and quantum cellular automata in one space dimension. In the classical case we are dealing with block cellular automata, whereas in the quantum case we switch to unitary quantum circuits of the brickwork type (quantum cellular automata). In this Section we review the basic constructions.

First we discuss the classical case. Let $X$ be a finite set with $N$ elements. We interpret $X$ as a local configuration space. A cellular automaton consists of a collection of cells and an update rule. We consider one dimensional systems, and we deal with classical "spin" variables $s_j$ with $j = 1, 2, \ldots, L$ that take values from the set $X$. Here $L$ is the length of the system, which is assumed to be an even number, and we will always consider periodic boundary conditions. A configuration $s = \{s_1, s_2, \ldots, s_L\}$ is an element of $X^L \equiv X \times X \times \cdots \times X$, and the update rule $\mathcal{V}$ is a map $X^L \to X^L$. This means that at each iteration we update the configuration as

$$s \quad \to \quad \mathcal{V}s. \tag{1}$$

For the identity map we will use the notation 1 throughout this work.

In this work we consider *information preserving* maps, which means that the update rule has to be an invertible map. The property of invertibility is analogous to "phase space conservation" in Hamiltonian mechanics, see the discussion in [51]. Furthermore we will also require time reflection invariance, which is an additional requirement; details will be specified below.

We consider Floquet-type block cellular automata. Specifically we restrict ourselves to two types of strictly local system, which we call $2 \to 2$ and $3 \to 1$ models. The Floquet rule consist of two steps in both cases: $\mathcal{V} = \mathcal{V}_2 \mathcal{V}_1$, and the maps $\mathcal{V}_{1,2}$ are constructed from commuting local maps. For convenience we choose the time coordinate $t$ such that the action of $\mathcal{V}$ corresponds to $t \to t + 2$.

In the case of the $2 \to 2$ models we deal with a local two-site map $U : X^2 \to X^2$ and build the update rules as

$$\begin{aligned}
\mathcal{V}_1 &= U_{L-1,L} \ldots U_{3,4} U_{1,2}, \\
\mathcal{V}_2 &= U_{L,1} \ldots U_{4,5} U_{2,3}.
\end{aligned} \tag{2}$$

We require time reflection invariance from our models in all cases. This means that the local update moves should be involutive, i.e. $U_{j,j+1}^2 = 1$. For the Floquet update rule this implies

$$\mathcal{V}^{-1} = (\mathcal{V}_2 \mathcal{V}_1)^{-1} = \mathcal{V}_1 \mathcal{V}_2. \tag{3}$$

In contrast to the $2 \to 2$ models the $3 \to 1$ models describe time evolution on light cone lattices, and each local update step uses the information from 3 sites to give a new value to a variable on one site. In most of this work we will focus on the $2 \to 2$ models, and we treat the $3 \to 1$ models in Section 6.

We also consider quantum circuits (also called quantum cellular automata, QCA), whose structure follows from an immediate generalization of what was discussed so far. In the quantum case we are dealing with local Hilbert spaces $\mathbb{C}^N$ whose basis is indexed by the elements of the finite set $X$. The full Hilbert space of a quantum chain of length $L$ is given by the $L$-fold tensor product of local spaces, and the state of the system is a vector $|\Psi\rangle$ of this Hilbert space.

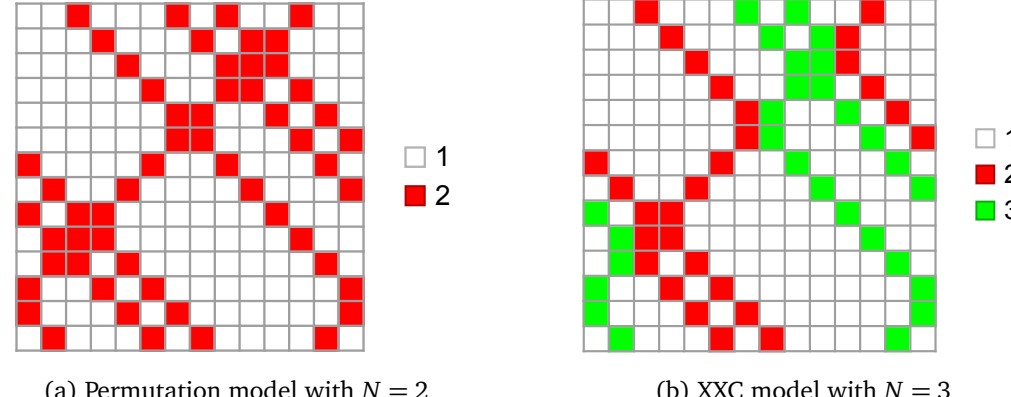

(a) Permutation model with $N = 2$          (b) XXC model with $N = 3$

Figure 1: Examples for time evolution in the classical cellular automata. In the plots the vertical direction corresponds to time, and the horizontal to space; the initial condition is given by the uppermost row, and time flows downwards. We plotted both half-steps of the Floquet cycle $\mathcal{V} = \mathcal{V}_2\mathcal{V}_1$, and we use a convention such that the addition of two rows corresponds to $t \to t + 2$. This way the speed of propagation in the permutation model is $v = \pm 1$. In the space direction periodic boundary conditions are applied.

In analogy with the classical case we are dealing with two-site maps $\hat{U}_{j,j+1}$ which are now quantum gates acting on a pair of local Hilbert spaces. Furthermore, we construct quantum circuits of brickwork type by the immediate generalization of the Floquet rule (2), by replacing classical maps with operators. Thus we obtain the quantum update rule $\hat{\mathcal{V}}$, and at each step the state of the system is changed as

$$|\Psi\rangle \quad \to \quad \hat{\mathcal{V}}|\Psi\rangle . \tag{4}$$

Every classical update rule can be lifted immediately to a quantum gate, by defining $\hat{U}$ such that it permutes pairs of basis vectors according to the classical map $U$. To be precise:

$$\hat{U}\left(|a\rangle \otimes |b\rangle\right) = |c\rangle \otimes |d\rangle , \text{ where } (c,d) = U(a,b). \tag{5}$$

In such a case the resulting quantum circuit is *deterministic* in the given basis. On the other hand, we will also consider cases where the two-site gate $\hat{U}_{j,j+1}$ produces linear combinations of the local product states, thus it becomes a true quantum gate.

## 2.1 Examples

Here we give two examples for the models that fit into our framework. Both models are classical, and they appeared earlier in the literature.

In the case of the classical $2 \to 2$ models the simplest one is what we call the permutation model. It is defined for every set $X$ by the permutation map $\mathcal{P}$:

$$U(a,b) = \mathcal{P}(a,b) \equiv (b,a), \qquad a,b \in X . \tag{6}$$

The application of the update rule leads to light cone propagation in the model, with the odd/even sub-lattices performing a cyclic shift to the right/left, respectively. An example for time evolution in this model is shown in Figure 1.

An other example is the XXC model with $N = 3$. Labelling the local configuration space as $X = \{1, 2, 3\}$ the update rule is defined as

$$U(a,b) = \begin{cases} (b,a) & \text{if } a = 1 \text{ or } b = 1, \\ (a,b) & \text{otherwise} . \end{cases} \tag{7}$$

This update rule is closely related to the XXC quantum spin chains studied in [52], hence the name. The map $U$ appears at the "rational point" of the $R$-matrices presented in [52]. The update rule was later independently proposed in [6] and the resulting model was further analyzed in [35,36]. In these works it was observed that the map $U$ satisfies the set theoretical Yang-Baxter relation (see Section 3).

In the XXC model the state 1 can be interpreted as the "vacuum", and the two remaining states can be interpreted as particles with an inner degree of freedom, given by two colors. An example for the time evolution in this model is shown in Fig. 1. If we focus only on the dynamics of the "charge", i.e. we forget about the colors, then the model is found to be equivalent to the permutation model. The new features come when we add the non-trivial color dynamics. It can be seen that the particle numbers for the two colors are separately conserved, but the colors do not propagate ballistically, the colors get reflected during scattering. Therefore, the spatial ordering of the colors is conserved during time evolution, while the particles move along the light cones. This color dynamics is typical for the low energy scattering in quantum models with two-color excitations. The model shows complex transport properties, and a number of exact results for the real time evolution were computed in [6,35,36].

## 2.2 Local symmetries and local conservation laws

We are interested in models that have a large set of local conservation laws, and sometimes also a finite set of local symmetries. Therefore we discuss these concepts briefly, and introduce the necessary notations. In these discussions we consider only the $2 \to 2$ maps, but the generalization to the $3 \to 1$ case is obvious.

First we discuss the symmetries and conservation laws in the classical language. The extension to the quantum case follows immediately, by lifting the classical maps to linear operators.

Let $S : X \to X$ be a bijection. We say that the model has global symmetry $S$, if

$$(S \times S)U_{1,2} = U_{1,2}(S \times S). \tag{8}$$

This implies that the global extension of $S$ defined as $S \times S \times \cdots \times S$ is a symmetry of the cellular automaton, it commutes with the global update rule $\mathcal{V}$.

It is also useful to discuss symmetries which can be applied locally. A special situation is if the symmetry operation can be applied along a light cone. We say that the symmetry operation $S$ propagates ballistically if

$$(1 \times S)U_{1,2} = U_{1,2}(S \times 1), \quad \text{and} \quad (S \times 1)U_{1,2} = U_{1,2}(1 \times S). \tag{9}$$

This implies that

$$\mathcal{V}S_{2k-1} = S_{2k+1}\mathcal{V}, \qquad \mathcal{V}S_{2k} = S_{2k-2}\mathcal{V}, \tag{10}$$

for every integer $k$, and we used the notation that $S_j$ acts on site $j$. Consecutive application of the relations above implies that the permutation symmetry "propagates along light cones" and it connects different orbits by changing the local states only along a specific light cone.

Let us also discuss conserved charges. We consider functions $q^{(\alpha)} : X^\alpha \to \mathbb{Z}$ with $\alpha = 1, 2, \dots$ and interpret them as local charge densities with range $\alpha$. We chose the image space of the charges to be the integers, which is very natural in systems with finite configuration spaces.

Correspondingly, we define translationally invariant extensive charges $Q^{(\alpha)} : X^L \to \mathbb{Z}$ as

$$Q^{(\alpha)} = \sum_{j=1}^{L} q^{(\alpha)}(j), \tag{11}$$

where it is understood that $q^{(\alpha)}(j)$ is the function $q^{(\alpha)}$ applied on the segment $[j, j+1, \ldots, j+\alpha-1]$ of the automaton, and periodic boundary conditions are understood. Our update rules $\mathcal{V}$ are invariant with respect to translation by two sites only, therefore it is useful to study charges with the same invariance properties. This leads to the combinations

$$Q^{(\alpha),+} = \sum_{j=1}^{L/2} q^{(\alpha)}(2j+1), \qquad Q^{(\alpha),-} = \sum_{j=1}^{L/2} q^{(\alpha)}(2j). \tag{12}$$

We say that a specific global charge $Q^{(\alpha)}$ is conserved, if

$$Q^{(\alpha)} = Q^{(\alpha)} \circ \mathcal{V} \tag{13}$$

in every volume $L \geq \alpha$, such that the charge density $q^{(\alpha)}$ is not changed as we increase $L$. The same definition applies for charges of the form (12).

Formulas (8)-(9) are immediately generalized to the quantum case. Regarding the charges, in the quantum case one deals with local operator densities $\hat{q}^{(\alpha)}$ and global charge operators $\hat{Q}^{(\alpha)}$ and the value of a given charge on a state $|\Psi\rangle$ is given by the mean value $\langle\Psi|\hat{Q}^{(\alpha)}|\Psi\rangle$. A quantum charge is conserved if it commutes with the time evolution as an operator:

$$\hat{Q}^{(\alpha)}\hat{\mathcal{V}} = \hat{\mathcal{V}}\hat{Q}^{(\alpha)}. \tag{14}$$

If the quantum circuit is deterministic, and if we restrict ourselves to states $|\Psi\rangle$ which are product states in our computational basis, then the statement (13) is equivalent to the conservation of $\langle\Psi|\hat{Q}^{(\alpha)}|\Psi\rangle$ which follows from (14). However, the commutativity (14) carries more information than the classical conservation law, which is stored in the off-diagonal elements of the relation.

In the quantum setting ballistically propagating one-site operators were studied in [53], they were called "solitons". In Clifford automata ballistically propagating multi-site charges were studied in [45] and they were called "gliders".

Let us now also define a concrete basis for the local charges in the classical setting. For one-site charges $X \to \mathbb{Z}$ let $[a]$ denote the characteristic function of the local state $a$. To be more precise the value of charge $[a]$ on a local variable $s \in X$ is

$$[a](s) = \begin{cases} 1 & \text{if } s = a, \\ 0 & \text{otherwise}. \end{cases} \tag{15}$$

This definition gives $N-1$ independent one-site charges, because $\sum_{a=1}^{N}[a] = 1$, where now 1 stands for the constant map $X \to \mathbb{Z}$ whose value is $1 \in \mathbb{Z}$.

Multi-site charges are then generated by products and sums of the one-site charges, and we use the notation $[a]_j$ for the one-site charge acting on site $j$. In the quantum mechanical setting the local operator corresponding to $[a]_j$ is the projector $P_j^a$ onto the local state $|a\rangle$ of site $j$. Combinations of such projectors span the vector space of the diagonal operators.

For a specific model let $N_r$ denote the number of independent extensive conserved charges with range $\alpha \leq r$. It is generally expected that an integrable cellular automaton should have an extensive set of conserved quantities, which means that $N_r$ should grow at least linearly with $r$. There are models where $N_r$ grows exponentially with $r$: we call these models super-integrable.

In analogy with the quantum case let us also define the "time evolution" for maps $f : X^L \to X^L$ in the classical case as

$$f \to \mathcal{V}^{-1} f \mathcal{V}. \tag{16}$$

This is a simple analog of the time evolution of quantum operators in the Heisenberg picture.

## 3 Cellular automata from Yang-Baxter maps

We consider cellular automata, where the update rules are solutions to the set theoretical Yang-Baxter equation. We will focus mainly on $2 \rightarrow 2$ models and we briefly discuss $3 \rightarrow 1$ models in Section 6. In the following Sections we focus mainly on the classical case, and we discuss the quantum extensions in 7.

The set theoretical Yang-Baxter equation was introduced by Drinfeld in [37]. It is a relation for maps $U : X^2 \rightarrow X^2$, which takes the form

$$U_{12}U_{23}U_{12} = U_{23}U_{12}U_{23}. \tag{17}$$

This is a relation for maps $X^3 \rightarrow X^3$, and it is understood that $U_{j,j+1}$ acts on the components $j, j+1$ of the triple product. Structurally it is equivalent to the so-called braid relation, or the "spectral parameter independent" quantum Yang-Baxter equation. Usually it is also required that $U^2 = 1$, although non-involutive solutions were also considered in the literature, see for example [39]. In this work we say that $U$ is a solution (or Yang-Baxter map) if it solves the relation (17) and it is involutive. We always restrict ourselves to space-reflection invariant cases, which is motivated by the physical applications. This requirement is usually not included in the definition of the Yang-Baxter map, but we do include it, so that we have reflection symmetry in both the space and time directions.

Every Yang-Baxter map $U$ gives rise to a solution of the usual (quantum) Yang-Baxter equation, this is treated later in 7. Yang-Baxter maps were studied in [38, 39] and in many other research works; we do not attempt to review the literature here. Concrete examples will be introduced later in Section 5.

Let us write the map $U$ as

$$U(x, y) = (F_x(y), G_y(x)), \tag{18}$$

where $F_x, G_x$ are maps $X \rightarrow X$ parameterized by an element $x$ of $X$. We say that the solution $U$ is non-degenerate if every map $F_x$ and $G_x$ is a bijection of $X$. The simplest example for a non-degenerate map is the permutation, when every $F_x$ and $G_x$ is the identity on $X$. In contrast, the identity solution given by

$$U(x, y) = (x, y) \tag{19}$$

is degenerate, because the functions $F_x$ map every $y$ to $x$.

Most of the literature focuses on non-degenerate maps due to their useful mathematical properties. Furthermore, in many papers the word "solution" is reserved to non-degenerate Yang-Baxter maps. However, for our purposes the degenerate maps are also important, so we do not exclude them from our studies. Recent papers which studied degenerate solutions include [54–57].

The non-degenerate property is a classical counterpart of the "dual unitarity" of the quantum gates. This is discussed in more details in Section 3.3.

### 3.1 Yang-Baxter maps and the permutation group

Here we discuss the implications of the Yang-Baxter relation. The ideas presented here appeared in many places in the literature, but the application to the block cellular automata seems to be new. Indeed it seems that the physical consequences that we find have not yet appeared in the literature.

The key idea is to relate the cellular automaton to the permutation group $S_L$ acting on $L$ elements. It is known that $S_L$ can be generated by the $L-1$ elementary permutations $\sigma_j$,

which exchange the sites $j$ and $j+1$. These generators are subject to relations

$$\sigma_j\sigma_{j+1}\sigma_j = \sigma_{j+1}\sigma_j\sigma_{j+1},$$
$$\sigma_j\sigma_k = \sigma_k\sigma_j, \text{ if } |j-k| \geq 2, \tag{20}$$
$$\sigma_j^2 = 1.$$

These relations uniquely determine the group $S_L$. Furthermore, they are formally equivalent to the relations satisfied by the local two-site maps $U_{j,j+1}$. Therefore we can generate a homomorphism $\Lambda$ from $S_L$ to bijective maps $X^L \to X^L$ by the assignments for the generators

$$\Lambda(\sigma_j) = U_{j,j+1}. \tag{21}$$

In other words the correspondence $\sigma_j \to U_{j,j+1}$ defines a group action of $S_L$ on $X^L$. The image of a generic element $\sigma \in S_L$ can be computed if we reconstruct it as a product of elementary exchanges. For example, the exchange of sites 1 and 3 is written as $\sigma_{1,3} = \sigma_{1,2}\sigma_{2,3}\sigma_{1,2}$ and this yields

$$\Lambda(\sigma_{1,3}) = U_{1,2}U_{2,3}U_{1,2}. \tag{22}$$

It is important that the homomorphism is not compatible with periodic boundary conditions, because we chose a specific set of elementary exchanges $\sigma_{1,2}, \sigma_{2,3}, \ldots, \sigma_{L-1,L}$ and the exchange of the first and last sites $\sigma_{1,L}$ is not a member of the generating set. In fact $\sigma_{1,L}$ can be expressed using the elementary generators as

$$\sigma_{1,L} = \sigma_1\sigma_2\ldots\sigma_{L-2}\sigma_{L-1}\sigma_{L-2}\ldots\sigma_2\sigma_1. \tag{23}$$

The corresponding map will be

$$\Lambda(\sigma_{1,L}) = U_{12}U_{23}\ldots U_{L-2,L-1}U_{L-1,L}U_{L-2,L-1}\ldots U_{23}U_{12}, \tag{24}$$

thus in the typical case

$$\Lambda(\sigma_{1,L}) \neq U_{1,L}. \tag{25}$$

Let us now define a simplified dynamical system directly for $S_L$; we call it the permutation system. The dynamical variable is now $\varsigma_t \in S_L$, where $t$ is the discrete time index, and the equation of motion is given by

$$\varsigma_{t+2} = V\varsigma_t, \tag{26}$$

where $V \in S_L$ is given explicitly by

$$V = \sigma_{1,L}\sigma_{L-2}\ldots\sigma_4\sigma_2\sigma_{L-1}\sigma_{L-3}\sigma_3\sigma_1. \tag{27}$$

Note that here every element is an elementary generator of $S_L$ except for $\sigma_{1,L}$.

This dynamical system is trivially solved: we find that after each iteration the elements on the odd/even sub-lattices move to the right/left by two sites. This implies $V^{L/2} = 1$, which means that the orbits close after $L/2$ iterations. Alternatively this can be seen if we write $V$ using permutation cycles as

$$V = (2, 4, 6, \ldots, L)(L-1, L-3, \ldots, 3, 1), \tag{28}$$

from which it is clear that the order of $V$ is $L/2$.

However, this does not imply the same behaviour for our permutation model. The update rule $\mathcal{V}$ of our cellular automaton is generally not compatible with the homomorphism:

$$\Lambda(V) \neq \mathcal{V}, \tag{29}$$

which follows simply from (25). Therefore the homomorphism can not be used to obtain global information about the cellular automata. Instead, it can give local information as discussed in the next Subsection. Essentially the same observations were already made in [12, 38]. The non-equality (29) allows for orbit lengths longer than $L$, except if the Yang-Baxter map is non-degenerate, see Subsection 3.3.

## 3.2 Conservation laws

Now we establish the existence of a large set of ballistically propagating maps (operators) and local charges. First we construct ballistically propagating operators in the permutation system, and then we project them down to the cellular automaton. As we described above, the time evolution of the permutation system is such that the even and odd sub-lattices are moving undisturbed to the left and to the right by two sites after each iteration. This means that every local operation or local charge which deals with only one sub-lattice will propagate freely to the left or to the right.

To be more precise let us choose a range $\alpha$ and let $\Sigma(j) \in S_L$ be a permutation which acts non-trivially only on the segment $[j, j+1, \ldots, j+2\alpha]$ such that it *only acts on the odd sites*, it leaves the elements on the even sub-lattice invariant, and $j \geq 1$ and $j + 2\alpha \leq L - 2$. Time evolution of the permutation system gives

$$V\Sigma(j)V^{-1} = \Sigma(j+2). \tag{30}$$

We interpret this as follows: under the adjoint action of $V$ the permutation is shifted to the right "ballistically", and there is no "operator spreading".

Now we apply the group homomorphism $\Lambda$ to the equation above. We assume that the segment $[j, j+1, \ldots, j+2\alpha]$ is well separated from the boundary sites $1$ and $L$, so that the boundary link $\sigma_{1,L}$ would not appear in the resulting computations. In this case we can use the homomorphism $\Lambda$ and we can actually make the substitution $\Lambda(V) \to \mathcal{V}$ for the time evolution of the localized operator, so that we obtain

$$\mathcal{V}\Lambda(\Sigma(j))\mathcal{V}^{-1} = \Lambda(\Sigma(j+2)). \tag{31}$$

This means that the operator $\Lambda(\Sigma(j))$ gets transformed ballistically on the cellular automaton, and there is no "operator spreading". Such operators were called gliders in [45].

Let us consider examples for this. The simplest permutation which acts only on odd sites is

$$\Sigma(1) = \sigma_{1,3} = \sigma_{1,2}\sigma_{2,3}\sigma_{1,2}. \tag{32}$$

Here we set $j = 1$ and $\alpha = 1$. The image of this operator under the homomorphism is

$$\Lambda(\Sigma(1)) = U_{12}U_{23}U_{12}. \tag{33}$$

Thus we obtain that the operator on the r.h.s. above propagates ballistically to the right. Actually in this simple case the relation

$$\mathcal{V}U_{12}U_{23}U_{12}\mathcal{V}^{-1} = U_{34}U_{45}U_{34} \tag{34}$$

can be established directly using the exchange relations (17) and the involutive property of the two site maps.

Ballistically propagating multi-site operators can be obtained in the same manner.

We can also construct ballistically propagating local charge densities. In this classical case we mimic the computation of mean values quantum mechanics. To every map $f : X^L \to X^L$ we associate a function $q : X^L \to \mathbb{Z}$ such that

$$q(s) = \begin{cases} 1 & \text{if } f(s) = s, \\ 0 & \text{otherwise}. \end{cases} \tag{35}$$

If the map $f$ acts only locally on a few sites, then the function $q$ will also depend only on the variables on those sites.

Let us now consider a local map $\Sigma(j)$ with range $2\alpha + 1$ which acts on the segment $[j, j + 1, \ldots, j + 2\alpha]$. This leads to a function $q(j)$ which depends only on the sites in $[j, j + 1, \ldots, j + 2\alpha]$. Then the equation of motion (31) implies that

$$q(j) \circ \mathcal{V} = q(j - 2). \tag{36}$$

This means that we obtained ballistically propagating charge densities. The chiral sums defined as

$$Q^+ = \sum_{j=1}^{L/2} q(2j + 1), \qquad Q^- = \sum_{j=1}^{L/2} q(2j), \tag{37}$$

are separately conserved, and it is important that the charge densities do not suffer "spreading". The simplest example is the three site charge obtained from (33). The concrete representation of these charges will depend on the model.

In the permutation system the number of such conserved operators grows as $(\alpha + 1)!$ with the range $2\alpha + 1$ as defined above. This growth appears faster than exponential. However, in a given cellular automaton with a finite $N$ there is only an exponential number of independent charges for a finite range $2\alpha + 1$. This means that the different charges that the construction gives will become linearly dependent as $\alpha$ is increased. The question of how many of them remain linearly independent can not be answered by our computation, and this can be model specific. Based on concrete examples it seems that there is always an exponential growth of the set of the independent charges, unless the model is completely trivial with $U = 1$. However, we can not prove this at the moment.

## 3.3 Dual unitary Yang-Baxter maps

An important implication of the non-degeneracy condition is, that in such cases there exists a "crossing" transformation for the map [38]. Let us view sets of elements $\{x, y, u, v\}$ as an allowed configuration if

$$R(x, y) = (u, v). \tag{38}$$

The non-degeneracy condition implies that the variables $x$ and $v$ can be "crossed", which means that for every pair $y$ and $v$ there is precisely one pair $x, u$ such that the relation (38) holds. This means that the update step is deterministic also when viewed as a "map acting in the space direction". It follows that the unitary operator $\hat{U}$ acting on $\mathbb{C}^N \times \mathbb{C}^N$ is a "dual unitary gate" [46–48]. A special property of such quantum circuits is that the infinite temperature one-point correlation functions are non-zero only along light cones, and within these rays one can observe ergodic, mixing, or stable behaviour, see [48]. Dual unitarity is a concept which is independent from integrability, although it was known that these properties can overlap. For local dimension $N = 2$ the only integrable dual unitary gates are equivalent to the permutation map multiplied by a diagonal unitary matrix [48]. Our contribution here is that the Yang-Baxter maps are integrable dual unitary models with local dimensions $N \geq 3$.

It was shown in [38] that a dual-unitary Yang-Baxter map induces a group action of $S_L$ on $X^L$ which is conjugate to the standard permutation action. To be more precise, it was explicitly shown that there exists a map $J_L : X^L \to X^L$ such that for every elementary exchange $\sigma_{j,j+1} \in S_L$

$$U_{j,j+1} = J_L^{-1} \mathcal{P}_{j,j+1} J_L, \tag{39}$$

where $\mathcal{P}_{j,j+1}$ is the operation in $S_L$ which exchanges the content of sites $j$ and $j + 1$. This homomorphism can be extended to the exchange of any two elements, for example

$$U_{1,L} = J_L^{-1} \mathcal{P}_{1,L} J_L, \tag{40}$$

which follows from (24) after substituting (39) for the elementary exchanges and then using the concrete algebra of the local permutation steps $\mathcal{P}_{j,k}$.

Combining these observations we get that in dual unitary models the homomorphism $\Lambda$ is actually compatible with the periodic boundary conditions, and the update rule of the automaton follows from

$$\mathcal{V} = J_L^{-1} \mathcal{V}_{\mathcal{P}} J_L \,, \tag{41}$$

where $\mathcal{V}_{\mathcal{P}}$ is the update rule of the permutation model.

The similarity transformation given by $J_L$ is typically quite non-local, thus the dynamics resulting from $\mathcal{V}$ can still be interesting from a physical point of view. Nevertheless an important consequence is that in dual unitary models

$$\mathcal{V}^{L/2} = 1 \,. \tag{42}$$

Thus the maximal length of the orbits is $L$ in these models.

We remark that the concept of "classical dual unitarity" is independent from integrability. In this work we consider only the Yang-Baxter maps, but other classical dual unitary models also deserve study, see also Section 8.

## 3.4 Commuting update rules

It is a central property of integrable models that there exist a large number of flows which commute with each other. In classical mechanics these flows are generated by the conserved functions and the Poisson bracket. In quantum mechanics the flows are generated by the Hermitian higher charges of the models. In contrast, such flows have not yet been discussed for the block cellular automata that we investigate. In the case of integrable quad equations the commuting flows are well understood, and postulating their existence can lead to a classification of such equations [10,58]. However, it appears that for BCA such flows have not yet been discussed.

Here we show how to compute different types of commuting update rules for our models, just by using the properties of the Yang-Baxter maps and the spatial ordering of the update steps. It would be desirable to obtain commuting update rules which are constructed in a similar way as our map $\mathcal{V}$ through (2) with some other two-site map $V_{j,j+1}$. However, it is not possible to find such maps without using concrete information about the model. Instead, we present commuting update rules with modified periodicity in space and time, such that the rules only use our basic map $U_{j,j+1}$. It is possible that in specific models additional commuting flows could be found on a case by case basis, but here we focus only on the generic structures. We take inspiration once again from the permutation system, whose dynamics is defined by (26). As explained above, the odd/even sites move to the right/left, in a uniform way. Therefore, in this system we can easily find update rules which commute with the main equation of motion: we can take any classical map which acts only on one of the sub-lattices. However, this comes at a cost: generally we need to break the translation symmetry of the chain in both the space and time directions. This is explained on the simplest example.

In $S_L$ with $L = 4k$ let us define the following two elements:

$$\begin{aligned} Z_1 &= \sigma_{L-3,L-1} \dots \sigma_{9,11} \sigma_{5,7} \sigma_{1,3} \,, \\ Z_2 &= \sigma_{L-1,1} \dots \sigma_{11,13} \sigma_{7,9} \sigma_{3,5} \,. \end{aligned} \tag{43}$$

The combination

$$Z = Z_2 Z_1 \tag{44}$$

generates a simple dynamics, where only sites on the odd sub-lattice are moved, and one half of them moves to the left, one half moves to the right. This update rule does not immediately

commute with $V$ defined in (27) above, instead we have

$$ZV^2 = V^2Z\,. \tag{45}$$

The update rule $Z$ is invariant with respect to translation with 4 sites, and (45) implies that within time period 4 it commutes with the dynamics of the actual model.

Now we use once again the homomorphism $\Lambda$ from $S_L$ to the maps $X^L \to X^L$, and we obtain a new update rule $\mathcal{Z} = \mathcal{Z}_2\mathcal{Z}_1$ for our cellular automaton, where we replace the permutations in (43) by their images under $\Lambda$. Earlier we computed (33), and now we use this to write

$$\begin{aligned}
\mathcal{Z}_1 &= \prod_{j=1}^{L/4} U_{4j+1,4j+2}U_{4j+2,4j+3}U_{4j+1,4j+2}\,, \\
\mathcal{Z}_2 &= \prod_{j=1}^{L/4} U_{4j-1,4j}U_{4j,4j+1}U_{4j-1,4j}\,.
\end{aligned} \tag{46}$$

In these formulas we included products that reach over the end of the spin chain. This is a consistent step, even though generally $\Lambda(\sigma_{1,L}) \neq U_{1,L}$. The reason why the manipulations work is that the commutation relations hold locally, and if we have products of non-overlapping operators, then the manipulations can be performed locally, without encountering problems regarding the global definition of the homomorphism $\Lambda$.

Note that these maps are just products of the operators $\Sigma$ defined in (33). We already established that the operators propagate ballistically on the chain, and this also implies that a non-overlapping product of them will be a "conserved map", which is equivalent to a commuting flow. The only complication is that commutation with a single instance of $\mathcal{V}$ would change the orders of $\mathcal{Z}_1$ and $\mathcal{Z}_2$ and that is why we need the commutation relation (45), which implies

$$\mathcal{Z}\mathcal{V}^2 = \mathcal{V}^2\mathcal{Z}\,. \tag{47}$$

The dynamics generated by $\mathcal{Z}$ is generally not trivial, this can be seen already in the permutation model.

For the sake of completeness we also a different type of commuting flow, where the locality of the new update rule is of different type as before. Let us construct the formal operator

$$\tilde{\mathcal{V}} = \ldots U_{3,4}U_{2,3}U_{1,2}\ldots\,. \tag{48}$$

We assume the product to be infinite in both directions. Such an operator is used in the construction of the box-ball systems [17], and in such a case the new value given to a variable depends on the equal-time values of the variables to the left of it. In this form the operator is not well defined due to the two boundaries at infinities. The usual way to deal with the problem is to assume that there is a vacuum configuration 0 such that it is an invariant global state of the map: $U(0,0) = (0,0)$, and we are dealing with configurations such that $s_j = 0$ for all $|j| > K$ with some large $K$. Then it is trivially shown that $\tilde{\mathcal{V}}$ commutes with our more standard update rule $\mathcal{V}$.

The problem with this definition is that in many models there can be different vacuum configurations, and depending on the choice of the vacuum at the two infinities we can get different action of $\tilde{\mathcal{V}}$ even in the bulk of the chain. In other words, this map is very sensitive to the boundary conditions, which always effect the bulk as well.

### 3.5 Open boundary conditions

For the sake of completeness we also consider cellular automata with open boundary conditions, and we focus specifically on free boundaries. In this case the Floquet update rule of the

automaton is $\mathcal{V}^B = \mathcal{V}_2^B \mathcal{V}_1$, where $\mathcal{V}_1$ is given by the same formula as in (2), but $\mathcal{V}_2^B$ is

$$\mathcal{V}_2^B = U_{L-2,L-1} \dots U_{4,5} U_{2,3}. \tag{49}$$

The only difference as opposed to $\mathcal{V}_2$ is that the boundary link $U_{L,1}$ is now missing.

This model has the same dynamics in the bulk as in the periodic, but the global dynamical properties are different. In this case the homomorphism from $S_L$ to the maps $X^L \to X^L$ works seamlessly, and we obtain

$$\mathcal{V}^B = \Lambda(V^B), \tag{50}$$

where $V^B \in S_L$ is given by the analogous formula

$$V^B = \sigma_{L-2} \dots \sigma_4 \sigma_2 \sigma_{L-1} \sigma_{L-2} \sigma_3 \sigma_1. \tag{51}$$

It is easily seen that this element of $S_L$ has order $L$, and it follows that

$$(\mathcal{V}^B)^L = 1. \tag{52}$$

Thus the cellular automaton with free boundaries has simpler global dynamics for all Yang-Baxter maps, and its orbit lengths are divisors of $L$. Writing $V^B$ using cycles we obtain

$$V^B = (2, 4, 6, \dots, L-2, L, L-1, L-3, \dots, 3, 1), \tag{53}$$

thus it is indeed an element of order $L$.

## 4 Conservation laws and ergodicity

In this Section we discuss the connection between the integrability and the ergodicity properties of the cellular automata. In a classical dynamical system ergodic behaviour means that the orbits cover the sub-manifold of the phase space allowed by the maximal set of conserved charges. Integrable systems have a lot of conservation laws, which implies that the orbits are restricted to sub-manifolds with much lower dimension than that of the full phase space. It is natural question to ask: How does this phenomenon manifest itself for classical cellular automata?

The configuration space of the finite volume cellular automata is finite, and it is a natural idea to study how this space splits into the orbits. In parallel we can ask: what is the maximal set of independent charges for a finite cellular automaton, and how does this relate to integrability?

Let us consider the orbits of configurations under time evolution generated by $\mathcal{V}$. The total configuration space splits into the union of orbits as $X^L = O_1 \cup O_2 \cup \dots \cup O_{N_o}$, where each $O_j$ is a closed orbit and $N_o$ denotes the total number of orbits. For each orbit $O_j$ we can define its characteristic function $Q^{(j)} : X^L \to \mathbb{Z}$ which takes the values

$$Q^{(j)}(s) = \begin{cases} 1 & \text{if } s \in O_j, \\ 0 & \text{if } s \notin O_j. \end{cases} \tag{54}$$

All of these functions are conserved by the time evolution, but they are not independent from each other. For example if we know that one of the characteristic functions takes value 1 on a specific configuration, then all of the other functions take value 0. In fact, for a finite automaton there is only one algebraically independent charge, which can be chosen as

$$Q = \sum_{j=1}^{N_o} j Q^{(j)}. \tag{55}$$

The value of this charge simply just tell us the index of the orbit to which the configuration belongs.

However, the construction of these charges involves the full solution of the time evolution, therefore they are not directly relevant for the discussions of integrability. The situation is similar to quantum mechanics, where in a finite Hilbert space a full set of conserved charges for a Hamiltonian $\hat{H}$ can be constructed using the projectors $\hat{P}_j = |j\rangle\langle j|$, where $|j\rangle$ are the eigenvectors of $\hat{H}$, but the existence of these operators involves the full solution of the model and therefore they do not tell us anything about the integrability properties.

One way out of this problem is to consider local charge densities as we did above. This generates extensive charges, whose construction is "stable" as the volume $L$ is increased. However, the drawback is that for such charges it is not immediately clear to what extent they foliate the finite configuration spaces. Generally we expect that if there are more and more functionally independent charges, then the foliation of the configuration space becomes more and more restrictive, leading to shorter and shorter orbit lengths. Let us now discuss three types of behaviour for this phenomenon.

A generic non-integrable model has a finite number of conserved local charges. In such models it is expected that the orbit lengths grow exponentially with the system size. In the most general case when there are no symmetries present (thus the model is not invariant with respect to space and time reflection, and there are no hidden symmetries either) we can expect that the orbit length grows in the same way as the configuration space grows, i.e. typical orbit lengths $\ell$ should scale as

$$\log(\ell) = L\log(N) + \dots, \tag{56}$$

where the sub-leading corrections grow slower than linearly in $L$. It is known that discrete symmetries can lead to less ergodic behaviour [59].

In a generic integrable model the number of local charges with range $\alpha$ grows linearly with $\alpha$. Therefore we expect that the total amount of information gathered from all the extensive local charges grows exponentially with the volume. Therefore we expect typical orbit lengths $\ell$ to scale as

$$\log(\ell) = Lc + \dots \quad , \text{where now} \quad 0 < c < \log(N). \tag{57}$$

In contrast, in superintegrable models we have an exponentially large set of local conservation laws. It is natural to expect that this will lead to sub-exponential growth of the orbit lengths. In particular, for the dual-unitary Yang-Baxter maps relation (42) states that the maximal orbit length is actually $L$, which is the same as for the permutation model. Then the question arises: What to expect from degenerate Yang-Baxter maps or other superintegrable cellular automata? For the superintegrable Rule54 model it was found in [25] that the orbit lengths grow quadratically with the volume. The same behaviour can be seen in the example of the XXC model with $N = 3$. Furthermore, for $N = 4$ we encountered models where the maximal orbit length grows as $L^3$, see below.

Motivated by the concrete examples (treated in the next Section) we propose to view the orbit lengths as a measure of the complexity of the super-integrable cellular automaton. We say that a specific model is of class $\mathcal{O}(L^m)$ if the maximal orbit length grows as $L^m$. Based on our concrete examples we conjecture that for every $m$ there is a super-integrable model of class $\mathcal{O}(L^m)$, with large enough $N$. Looking at the concrete examples we found that up to $N = 4$ all models in our class showed the expected polynomial growth.

At the same time we do not claim that every super-integrable model has polynomial growth. In fact if we relax some of our restrictions for the models, it is possible to find models with exponential growth as given by (57). For example for $N = 4$ we found a model which is not space-reflection symmetric, and which has exponential growth. However, in the this paper we constrain ourselves to models which are both space and time reflection invariant.

# 5 Constructions and examples

In this Section we discuss examples of models obtained from Yang-Baxter maps for various values of $N$. In all cases we consider only space reflection symmetric update rules. The identity map and the permutation map are the trivial solutions for every $N$, therefore we do not mention them separately.

For the small values $N = 2, 3, 4$ we performed a complete classification of involutive Yang-Baxter maps, considering both the degenerate and the non-degenerate cases. For non-degenerate maps a complete enumeration is available up to $N = 10$ in the work [39] and the associated github database. It is possible to extend the methods of [39] also to the degenerate cases, all solutions up to $N = 5$ were obtained this way, but this is unpublished [60]. For our own purposes we just applied a brute force search for solutions up to $N = 4$.

Before turning to the examples we discuss simple ways of constructing new automata from known ones.

Sometimes two different models can be connected by a site dependent twist transformation. Let $S$ be a local bijection, and let us consider the twist operation

$$\mathcal{S} = 1 \times S \times S^2 \times \cdots \times S^{L-1}. \tag{58}$$

Every $S$ is of finite order, and we assume for the moment that $L$ is a multiple of the order of $S$, thus the above map is compatible with periodic boundary conditions. We say that two different models defined by the maps $U_{1,2}$ and $\tilde{U}_{1,2}$ are related by a twist $S$, if $U$ and $\tilde{U}$ are symmetric with respect to $S$ in the sense of (8) and

$$U = (1 \times S)\tilde{U}(1 \times S^{-1}). \tag{59}$$

In such a case the global twist operator (58) connects the orbits of the two models with each other. Simple examples for such twists can be found if $S^2 = 1$, in which case

$$\mathcal{S} = 1 \times S \times 1 \times S \times \cdots \times 1 \times S \tag{60}$$

and thus we get a staggered similarity transformation between two models.

An other important construction is the "direct sum" of Yang-Baxter maps, with some allowed extra freedom. We say a Yang-Baxter map is decomposable if the set $X$ can be divided into two disjoint and non-empty sets $A$ and $B$ such that $U(A \times A) = A \times A$ and $U(B \times B) = B \times B$. In such a case the restrictions of $U$ to the subsets $A \times A$ and $B \times B$ have to be Yang-Baxter maps. We say that $U$ acting on $X = A \cup B$ is a simple union of two Yang-Baxter maps $U_A$ and $U_B$ if the restrictions to $A$ and $B$ are given by $U_A$ and $U_B$ and

$$U(a, b) = (b, a), \quad U(b, a) = (a, b), \qquad \text{for every } a \in A, b \in B. \tag{61}$$

Similar to the previous definition, we say that $U$ is a twisted union, if

$$U(a, b) = (f_a(b), a), \quad U(b, a) = (a, f_a(b)), \qquad \text{for every } a \in A, b \in B, \tag{62}$$

where $f_a$ is a map $B \to B$ parameterized by an element $a \in A$. The paper [38] also introduced generalized twisted unions, but we will not encounter them in our examples.

The permutation map on $X \times X$ is trivially decomposable: let $X = A \cup B$ such that $A$ and $B$ are non-empty and disjunct, then $\mathcal{P}$ for $X$ is a simple union of the permutation maps of $A$ and $B$. If $A$ or $B$ have more than one elements, they can be decomposed further. Eventually we see that the permutation map for $X$ is actually a simple union of single element sets.

The XXC models are examples for an other type of composition. Let us take two sets $A$ and $B$ and consider the identity maps on $A \times A$ and $B \times B$. Then the corresponding XXC model on

$X = A \cup B$ is the simple union of these identity maps [52,61]. The generalization to unions with more than two components appeared in [62]; the resulting systems were called "multiplicity models". For the sake of completeness we give the most general definition of the XXC or multiplicity models. Let us take a partitioning of the integer $N$ as $N = m_1 + m_2 + \cdots + m_n$ such that $m_j \leq m_k$ for $j < k$. Then we divide the set $X$ into subsets $A_j$ with cardinality $m_j$. The Yang-Baxter map is then given by

$$U(a,b) = \begin{cases} (b,a) & \text{if } a \in A_j, b \in A_k, \text{ and } j \neq k, \\ (a,b) & \text{if } a, b \in A_j. \end{cases} \tag{63}$$

For this map and the resulting model we will use the name XXC model of type $(m_1 + m_2 + \cdots + m_n)$. The update rule introduced in (7) is thus the XXC model of type $(1+2)$.

## 5.1 $N = 2$

For $N = 2$ there is only one model different from the identity and the permutation models. Using the finite group $\mathbb{Z}_2$ its update rule can be written as

$$U(a,b) = (b+1, a+1). \tag{64}$$

Using $X = \{0, 1\}$ the only non-trivial moves are

$$(0,0) \longleftrightarrow (1,1). \tag{65}$$

We can call it the spin-flip model. The map $U$ is globally symmetric with respect to the spin-flip $S$ given by $S(a) = a+1$. The model is related to the permutation model by an $S$-twist according to (60), which is essentially a spin flip performed on every second site. Alternatively, we can also write $U(a,b) = (S \times S)(b,a)$, which together with the $S$-symmetry implies that the update rule $\mathcal{V}$ of the model becomes identical to that of the permutation model. Therefore this is not a truly independent model, and it is trivially solved.

## 5.2 $N = 3$

For $N = 3$ we found a total number of 4 non-isomorphic non-trivial solutions. Out of the four there are only 2 models which can not be related to each other (or to the permutation model) by a twist.

We list the models below. We use the notation $X = \{1, 2, 3\}$. All non-trivial models are such that there is a distinguished element of $X$, and we choose this element to be 1. We interpret it as the "vacuum". Then the local states 2 and 3 are interpreted as an excitation with two colors. We will also use the color-flip transformation $S : X \to X$, which preserves the vacuum but flips the color of the excitation, thus it is now given by $S(1) = 1, S(2) = 3, S(3) = 2$.

- **Twisted permutation.** The model is given by

$$U(a,b) = (S(b), S(a)). \tag{66}$$

  The model is globally symmetric with respect to $S$, and it is related to the permutation model by the $S$-twist, which induces a color-flip on every second site. Alternatively, we can also write $U = (S \times S)\mathcal{P}$, which implies that the update rules is identical to that of the permutation model. Therefore the model is trivially solved.

- **Simple union of the vacuum state 1 and the spin-flip model acting on the states** {2, 3}**.** The non-trivial moves are

$$(2,1) \leftrightarrow (1,2), \qquad (3,1) \leftrightarrow (1,3), \qquad (2,2) \leftrightarrow (3,3). \qquad (67)$$

  The map is globally symmetric with respect to the color-flip $S$, which is actually a ballistically propagating symmetry. Two additive local charges are $[1]$ and $[2] + [3]$, and they are also ballistically propagating. Direct computation shows that $U$ is dual-unitary.

- **Twisted union of the vacuum state 1 and the permutation model acting on the states** {2, 3}**.** The non-trivial moves are now

$$(2,1) \leftrightarrow (1,3), \qquad (3,1) \leftrightarrow (1,2), \qquad (2,3) \leftrightarrow (3,2). \qquad (68)$$

  This model can be related to the previous one, if we perform a color-flip at every second site. It is also dual-unitary.

- **The XXC model of type (1+2).** The update rule of this model was given in eq. (7). This is the only non-trivial model with $N = 3$ which is not dual unitary. Additively conserved charges are $[1]$, $[2]$ and $[3]$, thus all particle numbers are conserved separately.

  The charges $[1]$ and $[2] + [3]$ are propagating ballistically. This means that every light cone is such that either it is always empty or it always has a particle, but in this case the color of the particle can change during time evolution. At the same time, the spatial ordering of the colors $[2]$ and $[3]$ is not changed during time evolution. Therefore we can trivially construct ballistically propagating multi-point charges: they are given by arbitrary products of $[1]$ and $[2] + [3]$ localized at the odd/even sub-lattice[1]. Our construction yields charges precisely of this type, for example the first ballistically propagating charge gives

$$U_{12}U_{23}U_{12} \quad \rightarrow \quad [1]_1[1]_3 + ([2]_1 + [3]_1)([2]_3 + [3]_3). \qquad (69)$$

  An example for time evolution for a random initial state (where the local states are chosen with equal probability) is shown in Fig. 2

  Regarding orbit lengths the XXC model of type $(1+2)$ is in the class of $\mathcal{O}(L^2)$ models. It can be seen that after $L$ steps the particle positions always return to their initial values, but the color arrangements (spatial ordering of the colors 2 and 3) can be shifted in either direction by some finite values. However, after $L^2$ steps the color arrangements also return to their initial configurations, thus the orbits close.

### 5.3   $N = 4$

We found a total number of 36 non-isomorphic non-trivial solutions. A number of them are related to each other by twists, and if we regard such models as equivalent then there remain 14 independent solutions. Out of these solutions 5 are dual unitary, and the remaining ones are non-trivial degenerate models.

   We do not list here all the 14 independent models, we just discuss some examples for them, focusing on models with special properties.

   First of all we list all models which preserve all the particle numbers separately. There are 3 non-trivial models, which are all of the XXC type, and they correspond to the different

---

[1]We acknowledge useful discussions with Tomaž Prosen about this question.

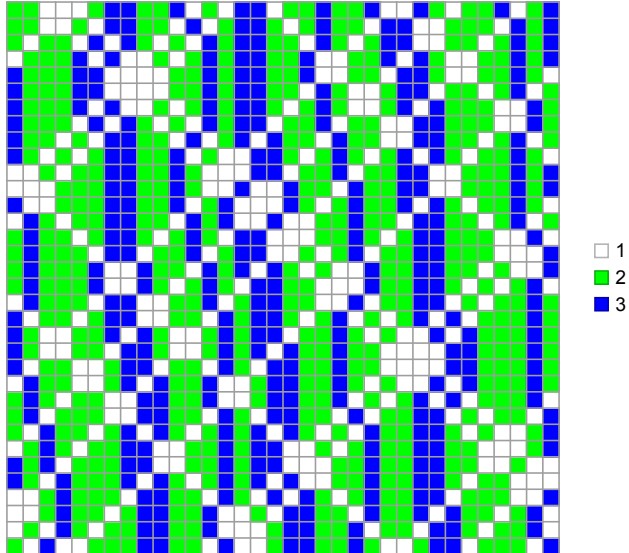

Figure 2: Example for time evolution in the XXC model of type $1+2$. The initial state is a random configuration with equal weights for the 3 states.

partitionings of the integer 4. To be concrete, they are the XXC models of type $(1+3)$, $(1+1+2)$ and $(2+2)$. The $(2+2)$ model will be discussed in more detail below.

Out of the 14 models there is just one linear map, which can be written using the finite group $\mathbb{Z}_4$ as

$$U(a, b) = (2a + b, 2b + a). \tag{70}$$

This model does not have any conserved one-site charges, and for $N = 4$ this is the only such model. It is dual unitary.

Regarding orbit lengths we observed that there are models of classes $\mathcal{O}(L)$, $\mathcal{O}(L^2)$ and $\mathcal{O}(L^3)$:

1. The models of class $\mathcal{O}(L)$ are the dual-unitary models, an essentially trivial model where the 4 states can be described by pairs of bits $\{a, b\}$, $a, b \in \mathbb{Z}_2$ and the update rule is

$$U(\{a, b\}, \{c, d\}) = (\{c + 1, b\}, \{a + 1, d\}) \tag{71}$$

and two additional models that are related to this one by twist transformations. The model given by (71) is trivially solved, because the dynamics of the first bits is given by the spin flip model (64), whereas the second bits are completely frozen.

2. We found 6 independent models of class $\mathcal{O}(L^2)$, among them are the XXC models of type $(1 + 3)$ and $(1 + 1 + 2)$. We do not discuss the remaining 4 models separately.

3. We found two independent models of class $\mathcal{O}(L^3)$. They are the XXC model of type $(2+2)$ and a similar model with a twisted union. These two models are discussed separately below.

### 5.3.1 The XXC model of type $(2 + 2)$

We analyze this specific model with more details, because it can have applications for the study of transport. For a concrete example of time evolution in this model see Fig. 3.

There are multiple ways of formulating the update rule. We can for example choose a decomposition $X = A \cup B$ with $A = \{1, 2\}$ and $B = \{3, 4\}$ and write

$$U(a, b) = \begin{cases} (a, b) & \text{if } a, b \in A \text{ or } a, b \in B, \\ (b, a) & \text{otherwise}. \end{cases} \tag{72}$$

Alternatively, the model can be seen as a specific discrete time analog of the Hubbard model. Let us consider particles with two possible spin orientations and the four local states $|\emptyset\rangle$, $|\uparrow\rangle$, $|\downarrow\rangle$, $|\uparrow\downarrow\rangle$. If we identify them with the states 1, 2, 3 and 4, respectively, then we obtain a model where particles can hop to neighbouring sites, but only the following hopping moves are allowed:

$$|\emptyset, \uparrow\rangle \leftrightarrow |\uparrow, \emptyset\rangle, \qquad |\emptyset, \downarrow\rangle \leftrightarrow |\downarrow, \emptyset\rangle, \qquad |\uparrow, \uparrow\downarrow\rangle \leftrightarrow |\uparrow\downarrow, \uparrow\rangle, \qquad |\downarrow, \uparrow\downarrow\rangle \leftrightarrow |\uparrow\downarrow, \downarrow\rangle. \tag{73}$$

One more rewriting of the update rule is the following. Let us represent the local states with a pair of bits $\{a, b\}$, such that the first bit encodes whether the local state is from the set $A$ or $B$, and the second bit tells us which state it is. The we can write

$$U(\{a, b\}, \{c, d\}) = \begin{cases} (\{c, b\}, \{a, d\}) & \text{if } a = c, \\ (\{c, d\}, \{a, b\}) & \text{if } a \neq c. \end{cases} \tag{74}$$

Notice that in this representation the first bit decouples from the second one. Therefore the update rule and the resulting dynamics can be seen as "nested", in the sense that the trivially computable orbits of the first bits control the information propagation on the second bit.

In the original formulation the additively conserved one-site charges of the model are $[1]$, $[2]$, $[3]$ and $[4]$, and the combinations $[1] + [2]$ and $[3] + [4]$ propagate ballistically. This corresponds to the decoupling of the first bits as explained above. The model is globally symmetric with respect to a finite permutation group generated by the exchanges $1 \leftrightarrow 2$, $3 \leftrightarrow 4$, and the combined exchange $1 \leftrightarrow 3$, $2 \leftrightarrow 4$.

An example for time evolution from a random initial condition is shown in Fig. 3.

Regarding orbit lengths the model is found to be of class $\mathcal{O}(L^3)$. A specific initial condition which leads to cubically increasing orbit lengths is if we consider a sequence consisting of a single 1, $2k + 1$ number of 2, a single 3, and $2k - 1$ number of 4, in the given order, such that $L = 4k$. It is easy to check that the corresponding orbit length becomes $2k(2k + 1)(2k - 1)$. Numerical investigation showed that there are no orbits that grow faster than $\mathcal{O}(L^3)$.

Finally we note that there is a different update rule of class $\mathcal{O}(L^3)$ which has some similarities with the XXC model of type $2 + 2$. It is a twisted union, and its non-trivial moves are

$$(1, 4) \leftrightarrow (4, 1), \qquad (1, 3) \leftrightarrow (3, 1), \qquad (4, 2) \leftrightarrow (2, 3), \qquad (3, 2) \leftrightarrow (2, 4). \tag{75}$$

We can see that now the scattering of the states 3 and 4 on 2 causes a color flip between 3 and 4. In this model the local one-site charges are $[1]$, $[2]$, $[3] + [4]$, and the combinations $[1] + [2]$ and $[3] + [4]$ propagate ballistically.

# 6 Extension to $3 \to 1$ models

In this Section we apply the previous ideas to the $3 \to 1$ models, that describe time evolution on light cone lattices. For a thorough introduction we recommend the papers [7, 8, 31].

We use the standard trick that we "double" the sites of the light cone lattice and then use the standard rectangular lattice as before. We deal with functions $u : X^3 \to X$ and build local update rules acting on three sites $U : X^3 \to X^3$ such that

$$U(l, d, r) = (l, u(l, d, r), r), \tag{76}$$

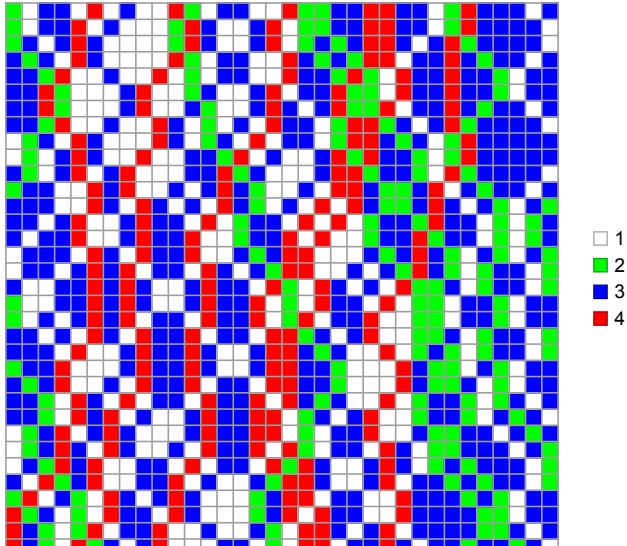

Figure 3: Example for time evolution in the XXC model of type $2+2$. The initial state is a random configuration with equal weights for the 4 states.

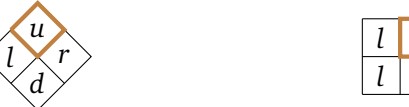

Figure 4: Update rule for $3 \rightarrow 1$ models. On the left the update is performed on the light cone lattice, such that the variable $u$ receives its value using the variables $l, d, r$. On the right the same update step is formulated on a rectangular lattice, in these case we deal with a 3-site rule, where the outer variables $l$ and $r$ act as control for the action on the middle variable $d$.

where $l, d, r$ are the input variables, from which $l$ and $r$ are control variables for the action on $d$. For a graphical interpretation of this update move see Fig. 4. The Floquet update rule is built essentially in the same way as in the case of the $2 \rightarrow 2$ models, now we have

$$
\begin{aligned}
\mathcal{V}_1 &= U_{L-1,L,1} \dots U_{3,4,5} U_{1,2,3} , \\
\mathcal{V}_2 &= U_{L,1,2} \dots U_{4,5,6} U_{2,3,4} .
\end{aligned}
\tag{77}
$$

The local update maps within each half-step still commute with each other, because their support overlaps only on control variables. In these models we also require time reversal symmetry, which amounts to $U^2_{j,j+1,j+2} = 1$.

Each $3 \rightarrow 1$ model can be described by a $2 \rightarrow 2$ model, if we perform a bond-site transformation. The idea is to put pairs of variables on the bonds (links) between sites, such that these variables completely describe the original configuration. In the most general case the bond variables can be chosen simply as a pair $(s_j, s_{j+1})$, where $s_j$ and $s_{j+1}$ are the variables on the two neighbouring sites which the bond connects. In this formulation the $3 \rightarrow 1$ map on three sites can be understood as a $2 \rightarrow 2$ map on two bonds as $((l,d),(d,r)) \rightarrow ((l,u),(u,r))$. Such a representation might not be economical, because the local dimension is increased to $N^2$. Nevertheless the bond-site transformation shows that there is no fundamental difference between the two types of models, and each integrable $3 \rightarrow 1$ model would be included in classifications of $2 \rightarrow 2$ models, if the local dimensions is chosen large enough. Sometimes in concrete cases more economic connections can be found, for example if the original model

has $\mathbb{Z}_N$ symmetry, in which case the bond-site transformation can be performed as

$$(s_j, s_{j+1}) \quad \rightarrow \quad s_j - s_{j+1}. \tag{78}$$

The simplest example for a $3 \rightarrow 1$ model with local dimension $N$ is a linear update rule using the additive group $\mathbb{Z}_N$. The update function is given by

$$u(l, r, d) = l + r - d. \tag{79}$$

This model is related directly to the permutation system, which is seen by performing the bond-site transformation mentioned above. Taking the three initial values $l, d, r$ and the three final values $l, u, r$, and computing the differences $(l - d, d - r)$ and $(l - u, u - r)$ we see that (79) is equivalent to a simple permutation in the bond model. Therefore, the dynamics can be understood using freely moving domain walls. In the specific case of $N = 2$ this model is the so-called Rule150 model [8], which was studied recently in [31, 63, 64].

Let us now generalize the notion of the Yang-Baxter map to the $3 \rightarrow 1$ models. The most natural generalization of the usual braid relation is the equation[2]

$$U_{123} U_{234} U_{123} = U_{234} U_{123} U_{234}, \tag{80}$$

which is a relation for maps $X^4 \rightarrow X^4$. The $3 \rightarrow 1$ maps automatically satisfy the condition

$$U_{j,j+1,j+2} U_{k,k+1,k+2} = U_{k,k+1,k+2} U_{j,j+1,j+2}, \text{ if } |j - k| \geq 2, \tag{81}$$

thus we obtain the same algebraic relations as in the $2 \rightarrow 2$ models. Such equations already appeared in [65], but it appears that for finite sets they have not yet been studied in detail.

It follows that all derivations presented in Section 3 follow through, with the obvious replacement

$$U_{j,j+1} \quad \rightarrow \quad U_{j,j+1,j+2} \tag{82}$$

in the computations. Therefore, $3 \rightarrow 1$ maps satisfying (80) can also be called Yang-Baxter maps, and they lead to superintegrable cellular automata.

We numerically investigated the solutions of (80) in the case of space reflection symmetric maps, and up to $N = 3$ we found that all of them are bond-site transformations of $2 \rightarrow 2$ models. However, this situation likely changes for higher $N$, and we expect that there are $3 \rightarrow 1$ models which can not be related to a $2 \rightarrow 2$ model with the same $N$.

One can raise questions about the dual-unitarity of such maps, and for generic, non-integrable $3 \rightarrow 1$ models this was investigated in [50]. A more detailed study of Yang-Baxter maps of the $3 \rightarrow 1$ type is beyond the scope of this paper.

# 7 Quantum circuits and related quantum spin chain models

In this Section let us turn to the quantum circuits and the related quantum spin chains. Once again we start with the $2 \rightarrow 2$ models.

Every Yang-Baxter map gives rise to a spectral parameter dependent solution of the quantum Yang-Baxter equation. Let $\check{R}(\lambda_1, \lambda_2)$ be the so-called $R$-matrix, which is an operator acting on $\mathbb{C}^N \otimes \mathbb{C}^N$ with two spectral parameters $\lambda_{1,2} \in \mathbb{C}$. The quantum Yang-Baxter relation reads

$$\check{R}_{12}(\lambda_2, \lambda_3) \check{R}_{23}(\lambda_1, \lambda_3) \check{R}_{12}(\lambda_1, \lambda_2) = \check{R}_{23}(\lambda_1, \lambda_2) \check{R}_{12}(\lambda_1, \lambda_3) \check{R}_{23}(\lambda_2, \lambda_3), \tag{83}$$

which is a relation for operators acting on $V \otimes V \otimes V$.

---

[2]We acknowledge very useful discussions with Vincent Pasquier about this and related equations.

The *R*-matrix corresponding to a Yang-Baxter map is simply

$$\check{R}_{j,k}(\lambda_j, \lambda_k) = \frac{1 + i(\lambda_j - \lambda_k)\hat{U}_{j,k}}{1 + i(\lambda_j - \lambda_k)},$$

(84)

where the linear operator $\hat{U}_{j,k} : V \otimes V \to V \otimes V$ is such that it permutes pairs of basis elements according to the action of the map $U_{j,k}$. The conventions above are chosen such that

$$\check{R}_{j,k}(\lambda)\check{R}_{j,k}(-\lambda) = 1.$$

(85)

Furthermore, if $\lambda \in \mathbb{R}$, then $\check{R}_{j,k}(\lambda)$ is unitary, which follows from the hermiticity of $\hat{U}_{j,k}$. It is easily verified that (84) solves (83) if the map $U$ is a Yang-Baxter map.

These *R*-matrices satisfy the so-called regularity property $\check{R}(0) = 1$, therefore they lead to integrable quantum spin chains with nearest neighbour Hamiltonians. Using the construction of [66] they also lead to integrable quantum circuits of the brickwork type.

Let us start with the discussion of the spin chains, where we are dealing with Hamiltonians that generate time evolution in continuous time. It can be shown using standard steps [67] that the resulting spin chain Hamiltonian is

$$\hat{H} = \sum_{j=1}^{L} \hat{U}_{j,j+1}$$

(86)

and it is integrable. Here periodic boundary conditions are understood. An extensive set of conserved charges can be derived from the usual transfer matrix construction.

The solution of these models is generally not known, but specific cases have been considered in the literature. For example the XXC-type Yang-Baxter maps lead to the models solved in [52, 62].

In the case of the dual unitary models the classical relation (39) was proven in [38]. This conjugation property carries over naturally to the quantum setting, and it implies that

$$\hat{U}_{j,j+1} = \hat{J}_L^{-1}\hat{\mathcal{P}}_{j,j+1}\hat{J}_L,$$

(87)

where now each linear operator acts on the Hilbert space. By the algebra of the permutation group we can extend this relation to any two-site exchange, and thus we obtain that the Hamiltonian above is conjugate to the fundamental $SU(N)$-symmetric model:

$$\hat{H} = \hat{J}_L^{-1}\left[\sum_{j=1}^{L}\hat{\mathcal{P}}_{j,j+1}\right]\hat{J}_L.$$

(88)

Periodic boundary conditions are understood also on the right. The operator $\hat{J}_L$ can be very non-local, but the connection says that the spectrum of the two Hamiltonians is the same. We stress that this holds only for the dual unitary models, but there it is true in both the periodic case and also for free boundary conditions. For the solution of the fundamental $SU(N)$-symmetric spin chain see for example [68].

In the case of the $3 \to 1$ models we need to use the formalism of the recent work [31] to generate quantum spin chains with medium range interactions.

If $U_{j,j+1,j+2}$ is a Yang-Baxter map in the sense of Section 6, then the corresponding translation invariant quantum spin chain is

$$\hat{H} = \sum_{j=1}^{L} \hat{U}_{j,j+1,j+2},$$

(89)

where the three site operator $\hat{U}_{j,j+1,j+2}$ follows directly from the three-site map $U_{j,j+1,j+2}$. These Hamiltonians are integrable. For such three site interacting models we need to consider a Lax operator that acts on the tensor product of three vector spaces. Using the formalism of [31] we find that these Lax operators are again linear in the spectral parameter:

$$\check{L}_{a,b,c}(\lambda) = \frac{1 + i\lambda \hat{U}_{a,b,c}}{1 + i\lambda} \,. \tag{90}$$

The integrability of the model is shown using the so-called GLL relation of [31], where the $\mathcal{G}$-matrix is chosen to be identical to the Lax operator.

These constructions are naturally extended to the quantum circuit setting [66]. The resulting brickwork circuits can be seen as "integrable Trotterizations" of the spin chains. Specifically, for $2 \to 2$ models the $R$-matrix (84) with some $\lambda \in \mathbb{R}$ can be used as a two-site quantum gate, and the resulting unitary circuit remains integrable [66]. In the case of the $3 \to 1$ models we can use the Lax operator (90) with some $\lambda \in \mathbb{R}$ as a three site quantum gate, and the resulting circuit remains integrable [31]. The notion of "Trotterization" comes from the fact that for small values of $\lambda$ the quantum update step $\hat{\mathcal{V}}$ can be seen as a discretization of the time evolution operator $e^{-i\hat{H}t}$ with $t = -\lambda$. The classical limit of the cellular automata are reproduced in the limit $\lambda \to \infty$ for both the $2 \to 2$ and $3 \to 3$ models, which is seen directly from (84) and (90). Therefore, a large (but not infinite) $\lambda$ corresponds to small quantum corrections on top of a classical time evolution.

It is important that in these models the spectral parameter $\lambda$ becomes a fixed parameter of the circuit, such that there is a set of conserved charges for each $\lambda$, but these sets of charges are not compatible with each other for different values $\lambda \neq \lambda'$. In accordance, the Hamiltonians (86) and (89) do not commute with the update rules of the quantum circuits. The smallest additively conserved charges of the quantum circuits (for generic values of $\lambda$) are three and four site charges, for $2 \to 2$ and $3 \to 1$ models, respectively, and the charges are derived from the transfer matrix constructions [31, 66].

The dynamics of the quantum circuits arising from Yang-Baxter maps have not yet been studied in the literature, except for the permutation map in the case $N = 2$, which leads to the Trotterization of the XXX Heisenberg spin chain [66]. Generally we expect that the phenomenology of the quantum circuits is much more rich than that of the classical cellular automata. This is already seen in the simplest case of the permutation map [66]. The superintegrability of the systems holds only in the classical limit: most of the charges of the classical automata cease to be conserved in the quantum setting.

If the classical Yang-Baxter map was dual-unitary, then this property is lost in the integrable quantum circuits. This is a simple consequence of the formulas (84) and (90): the identity operator singles out the time direction, and this is not compatible with the idea of dual unitarity. For dual unitary deformations of dual unitary maps see the next Section.

## 8 Non-integrable dual unitary gates

In those cases when the classical YB map is dual unitary, it is possible to continuously deform the resulting quantum circuits by keeping the dual unitarity. In this process the integrability is generally lost, which means that the resulting models will not have an infinite set of local conserved charges.

The construction is very similar to what was proposed in [48] and more recently in [49]; for generic non-integrable dual unitary gates the same ideas appeared in [69]. We use the classical Yang-Baxter map as the "core" of the dual unitary gate, which is then dressed with phases and external single site unitaries. The formula for a dressed dual unitary gate $\hat{V}_{1,2}$

acting on sites 1 and 2 reads

$$\hat{V}_{1,2} = B_1^- B_2^+ \hat{J}_{1,2} \hat{U}_{1,2} A_1^+ A_2^-,  \tag{91}$$

where $\hat{U}_{1,2}$ is the deterministic linear operator obtained from the Yang-Baxter map, $A^\pm, B^\pm \in SU(N)$ are single site unitaries, and $\hat{J}_{1,2}$ is a diagonal matrix in the computational basis whose matrix elements are pure phases. It follows from the deterministic nature of $\hat{U}_{1,2}$ that the product $\hat{J}_{1,2}\hat{U}_{1,2}$ is also dual-unitary. The physics of the quantum circuit is affected only by the combinations $A^+ B^+$ and $A^- B^-$, thus we are free to set for example $A^\pm = 1$.

We can see these quantum gates as deformations of the super-integrable cellular automata. However, the integrability of the classical update rule $U_{1,2}$ is not used in the parameterization (91), the only important piece of information is the non-degeneracy (or classical dual-unitarity) of the map.

## 9 Discussion

Let us summarize here the main results of this work. Starting from Yang-Baxter maps we constructed super-integrable classical cellular automata, and showed the existence of an exponentially large set of local conserved charges. These are such that the charge densities propagate ballistically on the chain, without "operator spreading". One could argue that the presence of such charges makes the dynamics too trivial, but this is not the case. Many such models possess additional conserved charges on top of the ballistically propagating ones, and the transport of these quantities can show typical behaviour of more generic systems, for example co-existence of ballistic and diffusive transport, as shown first in [6].

A central result of our work is that the so-called non-degenerate maps lead to a classical version of the dual unitary quantum gates. These models are the most constrained ones: using the results of [38] we showed that the dynamics is equivalent to that of the permutation model, and the equivalence is given by a known similarity transformation. Thus these models can be considered as "solved" from a mathematical point of view. However, there could be still interesting dynamics from a physical point of view, because the similarity transformation is highly non-local.

We characterized the dynamical complexity of the models by looking at the orbit lengths in finite volume, and we observed that in most models the maximal orbit length grows polynomially with the volume. This was found to hold for all space reflection symmetric models up to $N = 4$, and counter-examples were found only if space-reflection invariance is broken. Once again the dual unitary models were found to be the most constrained, where the maximal orbit length is always $L$.

While we showed the superintegrability of the models, we did not provide exact solutions for the time dependence of physical observables. We believe that these models are exactly solvable, but whether there is a general strategy for the computation of observables of interest, or whether it has to be done on a case by case basis, remains to be seen. The XXC model of type 2+2 (discussed in Section 5.3.1) seems a good candidate for further studies, because it can be seen as a Hubbard-like classical automaton, and also as a highly symmetric toy model for diffusive transport.

As a by-product of our computations we encountered an interesting family of dual unitary quantum gates, given in Section 8, studied earlier for generic non-integrable cases in [69]. It would be interesting to extend these ideas to the hexagonal geometry discussed in [70].

In Appendix A below we show that not all integrable BCA originate from Yang-Baxter maps. This demonstrates that the world of integrable BCA is quite rich and it deserves further study.

## Acknowledgments

We are thankful to Tomaž Prosen, Vincent Pasquier, Sarang Gopalakrishnan, Andrew Kels, Romain Vasseur, Leandro Vendramin and Levente Pristyák for useful discussions.

## A    Integrable cellular automata that do not come from Yang-Baxter maps

Here we give examples for models which appear to be integrable such that the update rule does not satisfy the Yang-Baxter equation. We consider models that fit into the framework used in this paper, and all our examples are $3 \to 1$ models.

First of all we mention the celebrated Rule54 model [7, 8]. Its update function can be formulated using the finite field $X = \mathbb{Z}_2$ as

$$u(l, r, d) = d + l + r + lr \, . \tag{92}$$

It is known that the model is super-integrable, however, the algebraic origins of its integrability are still not clear [7]. Direct substitution into (80) shows that the map is not a Yang-Baxter map. In [30] an attempt was made to capture the underlying integrability using the Yang-Baxter equation in the interaction-round-a-face (IRF) form, but it was shown in [31] that this approach does not give additional conserved charges for the model. There exists a quantum deformation of the Rule54 model, which involves a dynamics generating six-site charge that commutes with the original update rule [71]. However, the full Yang-Baxter integrability of the model was not yet worked out. We should note that the equation (92) appeared in the classification [58] of classical integrable quad equations, however, that work concerns equations over the complex numbers.

More recently an integrable 2-color extension of the Rule54 model was proposed in [72]. Let us choose now $X = \{0, 1, 2\}$. The local state 0 is interpreted as the vacuum, and the states 1 and 2 describe excitations with two colors. The update rule can be written using the finite field $\mathbb{Z}_3$ in the compact form

$$u = (l + r)(l^2 + r^2) + d(1 + (l^2 + r^2)^2) \, . \tag{93}$$

This form of the update function is the most compact representation, but its physical meaning is not transparent. In [72] the same function was presented on a case by case basis, with more physical insight.

In this model the particle numbers of the two species are not conserved separately, just a specific combination is conserved [72]. It was found in [72] that the model is integrable: it appears to have an infinite set of local charges such that the number of the charges grows linearly with the range. However, the algebraic origin of its integrability is not clear at the moment. Direct substitution into (80) shows that the map is not a Yang-Baxter map. This is already clear from the fact that it is an extension of the Rule54 model, whose update rule is not a Yang-Baxter map.

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
