# Peer review of "Superintegrable cellular automata and dual unitary gates from Yang-Baxter maps"

_SciPost Physics, doi:SciPost Phys. 12, 102 (2022)_

## Round 2 · Referee Report · Pieter W. Claeys (Referee 1) · 2022-2-16

Strengths

-Detailed analysis of the dynamics in cellular automata with local updates given by Yang-Baxter maps. - Illustrative examples with nontrivial physics. - New classes of cellular automata and dual-unitary gates are presented.

Weaknesses

  • Consequences of classical dual-unitarity are only briefly discussed.
  • Section 9, while interesting, is largely unrelated to the bulk of the paper.

Report

In this work the authors investigate cellular automata where the local update rules are given by Yang-Baxter maps satisfying the braid relation. For such cellular automata the local update rules can be mapped to elementary permutations and the resulting dynamics can be analyzed for the corresponding 'permutation system'. While this homomorphism can not be used to extract global information about the dynamics due to its breakdown at (periodic) boundaries, local information can be directly obtained starting from this simpler permutation system. The authors use this mapping to illustrate a lack of "operator spreading" and construct ballistically spreading local charge densities as well as commuting update rules. After a discussion of the notions of ergodicity/complexity and orbit lengths in classical cellular automata the authors discuss various constructions and present several illustrative and nontrivial examples. These examples are superintegrable and the orbit lengths are classified and shown to grow polynomially with system size. The authors also make clear which properties are expected for general Yang-Baxter cellular automata and which are specific to the presented examples. While the bulk of the paper focuses on classical cellular automata, it is also shown how Yang-Baxter maps give rise to quantum circuits acting as "integrable Trotterizations" of the corresponding integrable quantum spin chains.

A second aim of this paper is to relate the classical Yang-Baxter maps to (quantum) dual-unitary gates. It is shown that 'non-degenerate' Yang-Baxter maps can be used to construct classical dual-unitary gates, for which the mapping to the permutation system is compatible with the periodic boundary conditions — leading to a maximal orbital length that equals the system size. Furthermore, a new class of quantum dual-unitary gates is constructed by an appropriate dressing of the classical gate.

As pointed out by the authors, the idea of combining Yang-Baxter maps with cellular automata is natural and relates to various topics of interest. The presented analysis and results are technically sound and sure to be of interest to anyone working in related fields. Furthermore, the paper is written in an extremely clear and accessible way, including various illustrative and pedagogical examples. While various ideas in this work already appeared in the literature, the authors consistently make clear which ideas are new and provide appropriate references and context otherwise.

Considering SciPost Physics' acceptance criteria, I believe this paper satisfies criteria 3) Open a new pathway in an existing or a new research direction, with clear potential for multipronged follow-up work; and 4) Provide a novel and synergetic link between different research areas. Criterion 3) is evidenced by the new cellular automata introduced in the second half of the paper, and criterion 4) by the aim of the paper to combine Yang-Baxter maps with cellular automata.

I very much enjoyed reading this paper and am happy to recommend it for publication in SciPost Physics. Some minor comments can be found below, but these are only meant for clarification and are optional.

Requested changes

As mentioned in my report, these are mainly meant for clarification.

1- In Section 3.3 it is argued that non-degenerate Yang-Baxter maps give rise to maps that are also deterministic when acting in the space direction — the classical equivalent of dual-unitarity. Does the map acting in the space direction, i.e. the 'dual' of $U$, also satisfy the Yang-Baxter equation? There also exist dual-unitary gates parametrized in terms of Latin squares, as discussed by one of the authors in another work. Are these related to the non-degenerate Yang-Baxter maps? It also seems as if the 'non-degeneracy' suffices to return a dual-unitary map, even without the Yang-Baxter properties, but then the resulting dynamics no longer map to the permutation system. Can the authors comment on this? This would help to clarify the difference between the physical consequences of 'classical dual-unitarity' and 'classical dual-unitarity for Yang-Baxter maps'.

As a minor detail, in Eq. (38) the notation $R(x,y)$ is introduced, whereas otherwise the authors use $U(x,y)$.

2- When introducing the permutation system in Eq. (27), it might be useful to recall the initial definition of $\mathcal{V}=\mathcal{V}_2\mathcal{V}_1$. Even if $\Lambda(V)\neq \mathcal{V}$, this makes the motivation for Eq. (27) more explicit, which was initially not clear to me.

3- In Section 3.3. it is discussed how classical dual-unitary maps can be mapped to the permutation system. Does this property relate to the 'exact solvability' of dual-unitary gates in the quantum case? Is there any trace of the dual-unitarity light-cone dynamics of one-point correlation functions in the classical case?

---

## Round 2 · Referee Report · Anonymous (Referee 2) · 2022-2-21

Report

The authors study deterministic circuits that arise from Yang-Baxter maps: set theoretical solutions of the Yang-Baxter equation. They show that this structure implies the existence of ballistically propagating local operators that experience no broadening, which in turn imply a super-exponentially-growing number of local conservation laws, of which an exponential number appears to be linearly independent. This is in contrast with usual integrable systems, where conservation laws are expected to grow linearly with their support. The authors also discuss the ergodicity properties of these systems, and the existence of higher commuting update rules. The next part of the paper is mostly devoted to constructions and examples of these cellular automata. Out of the full classification of integrable Yang-Baxter maps up to local configuration space N=4, they pick a few representative examples, and discuss the obvious mapping between the models. The authors also briefly show how the previous ideas can be applied to 3->1 models, and to quantum models. They conclude with a brief discussion of non-integrable dual-unitary gates.

The paper presents a collection of interesting results and represents an impressive step towards classification and construction of solvable circuits, which have been in the past mostly studied on the case-by-case basis. Furthermore, the authors bring some more specialised mathematical literature to the attention of the wider community, and the discussion is clearly structured, which makes the article easy to follow. Therefore I strongly recommend this paper for publication in SciPost. In particular, I believe this article matches points 3 and 4 of the journal's acceptance criteria.

Requested changes

I have a few minor remarks: 1- Line after eq. (39): If I understand correctly $P_{j,j+1}$ is the operation in $X^L$ and not $S_L$. 2- Line before eq. (74): "The we" -> "Then we" 3- Sec. 7: In a few places $\mathcal{V}$ is used instead of (what I think should be) $\mathbb{C}^N$, e.g. first line after eq. (83) and (84). 4- Paragraph after eq. (88), "We stress that this holds...": I suggest to make this sentence a bit clearer also for a more sloppy reader. Maybe removing "there" could help. To me it was not immediately clear that "but there..." refers to dual unitary models (i.e. the cases in which it holds).

---

## Editorial Decision

published